# Can Vibrational Playbacks Disrupt Mating or Influence Other Relevant Behaviours in *Bactericera cockerelli* (Triozidae: Hemiptera)?

**DOI:** 10.3390/insects11050299

**Published:** 2020-05-12

**Authors:** Sabina Avosani, Thomas E. Sullivan, Marco Ciolli, Valerio Mazzoni, David Maxwell Suckling

**Affiliations:** 1DICAM Department of Civil, Environmental and Mechanical Engineering, University of Trento, 38123 Trento, Italy; marco.ciolli@unitn.it; 2Research and Innovation Center, Fondazione Edmund Mach, I-38010 San Michele all’Adige, Italy; valerio.mazzoni@fmach.it; 3The New Zealand Institute for Plant and Food Research Ltd., Lincoln, Christchurch PB 4704, New Zealand; Thomas.Sullivan@plantandfood.co.nz (T.E.S.); Max.suckling@plantandfood.co.nz (D.M.S.); 4Center of Agriculture, Food and Environment (C3A), University of Trento, I-38010 San Michele all’Adige, Italy; 5Centre for Biodiversity and Biosecurity, School of Biological Sciences, University of Auckland, PB 92019, Auckland 1142, New Zealand

**Keywords:** behavioural manipulation, vibrational signals, tomato potato psyllid, mating disruption, pest control

## Abstract

Behaviours of insects can be manipulated by transmitting vibrational signals to host plants in order to develop pest management techniques. *Bactericera cockerelli* is an important pest and uses vibrations for mate-finding. In order to design a future control strategy for *B. cockerelli*, three different bioassays were performed to assess whether vibrational signals could affect relevant behaviours. Single males or pairs were treated with a female playback in test 1 and 2, respectively. In test 3, mixed sex groups received either different disturbance playbacks. The use of a female playback significantly reduced the mating success of males, since they were attracted towards the source of the stimulus. Moreover, test 2 revealed that *B. cockerelli* females are competitive, since they used their signals to cover the playback and to duet with males, while in test 3, the disturbance playback, consisting of broadband noises significantly reduced male signalling activity. However, none of the treatments of test 3 negatively affected the mating success of males, which tended to mount the other conspecifics present on the same leaf. The role of vibrations in sexual communication and their potential application as control technique for *B. cockerelli* are discussed as well.

## 1. Introduction

Many insect species rely on vibrational signals to interact with co- and heterospecifics in different behavioural contexts [1,2]. Recent studies of applied biotremology demonstrated that vibrations can be used to either manipulate or disrupt insect behaviours in order to control pests’ populations [3,4]. Vibrational techniques are eco-friendly control practices that could be applied in the field to support integrated pest management strategies and decrease the use of pesticides. For instance, the mating behaviour of the leafhopper *Scaphoideus titanus* was successfully disrupted in semi-field conditions by the transmission of the male rivalry signal into grapevine [5], leading to the realization of the first vibrational vineyard in Italy for the control of grapevine leafhoppers since 2017 [6]. Another example of the use of vibrations for pest control is the transmission of a female Asian citrus psyllid *Diaphorina citri* playback to small citrus trees to disrupt mating and attract males toward the vibrating traps [7,8]. Other applications, currently objects of research involve the glassy-winged sharpshooter *Homalodisca vitripennis* [9,10], and the brown marmorated stinkbug *Halyomorpha halys* [11,12].

In this research work, we explored the use of vibrational signals to manipulate the mating behaviour of the tomato potato psyllid *Bactericera cockerelli*. This is an important pest of solanaceous crops such as potato (*Solanum tuberosum*), tomato (*Solanum lycopersicum*) and pepper (*Capsicum annum*) [13,14]. Its infestations resulted in severe losses in the United States, Mexico, Central America and New Zealand [13,14,15,16]. Besides damage due to direct feeding, the psyllid can transmit the bacterium *Candidatus* Liberibacter solanacearum [17,18], which causes a severe disease known as ‘’zebra chip’’ in commercial potatoes [18,19]. Current control strategies are relatively expensive and mainly consist of using pesticides, hence ongoing research is addressed to design novel approaches for the management of this pest [14]. Alternative techniques could involve the application of vibrations to affect the psyllids’ mating behaviours, since they are commonly mediated by vibrational signals [20,21,22]. Similarly to other psyllid species [20], the sexual behaviour of *B. cockerelli* depends upon the establishment of a vibrational duet between a female and a male, in which the latter actively searches for the stationary female on the plant [23]. Although the male is usually the first to emit a calling signal (the male calling signal, MCS), the female could advertise her presence with the emission of a female attraction signal (FAS), while the female response signal (FRS) is used to duet with the male. The FAS and the FRS have been used to synthesize a female playback, which was already tested on single males placed on a capsicum leaf. It was found that the transmission of the playback to the leaf, not only triggered the male signalling activity, but also drove the psyllid towards the source of the stimulus (i.e., a mini shaker; [23]).

Besides natural pre-recorded signals, intense sounds can affect insect behaviour, as demonstrated for the green peach aphid *Myzus persicae* [24]. In this case, sound stimuli with frequencies between 100 and 10,000 Hz transmitted at 66- and 90-dB SPL suppressed the feeding activity of the aphids [24]. Electronic music was recently used to disrupt insect behaviours as well. The airborne transmission of the song *Scary Monsters and Nice Sprites* by Skrillex [25], for instance, negatively influenced feeding, frequency of host attack, and mating in the mosquito *Aedes aegypti* [26]. However, *B. cockerelli* uses vibrational signals to communicate, thus airborne sounds are likely less effective than vibrations in disrupting relevant behaviours.

Therefore, we performed three experiments using vibrational stimuli transmitted to the plant tissues with the aim of affecting the mating behaviour and/or the persistence of psyllids on the host. In test 1 and 2, we used a female playback that was 15 s longer than the stimulus used in the previous work of Avosani and colleagues [23]. We hypothesised that a longer signal could be more efficient in attracting the male toward the source, probably by providing more directional information [27]. In test 1, the playback was tested on single males to evaluate its reliability in mimicking a real female and leading the insect towards the source of the stimulus. The same playback was then applied on pairs (a female and a male) in test 2 as an experiment of mating disruption. In this case, we assessed whether the playback transmission affected or not the mating success. Finally, in test 3, we investigated if the transmission of other unspecific vibrational signals (i.e., broadband noises and electronic music) to plant tissues could affect some aspects of the psyllid behaviour (i.e., signalling activity, mating success and persistence on the plant).

## 2. Materials and Methods

### 2.1. Insects’ Rearing

A colony of *B. cockerelli* was kept under controlled conditions (23 ± 2 °C, 60 ± 5% RH, 16:8 L:D) in a climate chamber at Plant and Food Research in Lincoln (Canterbury, New Zealand). The insects were maintained on capsicum (*Capsicum annuum*) inside rearing cages (Bugdorm-6620, MegaView Science Co., Ltd., Taiwan; 60 cm × 60 cm × 120 cm).

### 2.2. Recordings

The arena used in our trials consisted of a capsicum cutting composed of a stem and either two or three leaves (Figure 1) depending on the test. The cutting was placed in a vial filled with water to prevent withering, while a laser vibrometer (PDV 100, Polytec, Waldbrann, Germany) was pointed onto a piece (0.5 × 0.5 cm) of reflective tape glued to a leaf; the recorded signals were digitized and stored via a sound card (M-Audio Fast Track Pro, M-Audio, UK) at a 48 kHz sample rate and 24-bit resolution on a laptop computer (HP, EliteBook 8460 p) using Raven Pro 1.6 (The Cornell Lab of Ornithology, Ithaca, NY, USA). The laser was set on a steel plate (1.2m × 1.2m × 10 mm; Slade Engineering, Christchurch, New Zealand), which was placed on an array of 9 insulating rubber pads over a table. To associate the behaviour and respective emission of vibrational signals, insect movements were continuously followed using two cameras (Logitech Quickcam Pro 9000 webcam, Logitech International S.A., Lausanne, Switzerland) (Microsoft^®^ LifeCam HD-5000, Microsoft Corporation, Albuquerque, NM, USA), pointed on the ad-axial and ab-axial sides of the leaf, respectively. The software Bandicam 4.5. (Bandicam Company 2008–2020, Irvine, CA, USA) was used to control the cameras to follow the movement of the psyllids during the trials. Vibrations and insect behaviours were recorded throughout all the playback trials, in order to evaluate the emission of signals by the psyllids and to verify the correct transmission of the playbacks.

### 2.3. Playback Trials

The trials were carried out from December 2019 to January 2020 at Plant and Food Research in Lincoln (Canterbury, New Zealand). The playbacks were transmitted to the substrate using an electromagnetic mini shaker (Direct Drive Linear Motor, model DDLM-038-051-01, MotiCont, Los Angeles, CA, USA) in direct contact with the stem of the cutting (Figure 1). A sharpened steel rod was screwed at the top of the device, which was set on a nearby table using a clamp. In total three experiments were conducted at room temperature (25 ± 2 °C).

Using a procedure similar to an experiment carried out in [23], pre-recorded psyllid signals were used to assemble a female playback consisting of a female attraction signal (FAS) followed by a female response signal (FRS) after 25.05 s of silence. The signals were 27.7 s and 53.3 s long, respectively. The repetition time between pulses, the interval between pulse trains, and the frequency ranges of the playback were the same as the natural pre-recorded signals. The intensity of the female playback was adjusted to not exceed the insect natural amplitude. The duration of the FAS was not modified, while the FRS duration was 15 s longer than the pre-recorded signal (the average duration of the FRS is 42.8 ± 19.4 s; [23]). Although *B. cockerelli* is a polygamous insect [28], psyllids were removed from the mass colony and separated by gender two days before performing the test, in order to increase their motivation to mating.

#### 2.3.1. Test 1. Female Playback on Single Males

The female playback was tested on single males (*n* = 20) as a loop for 20 min to assess whether the stimulus could trigger the male search and lead the psyllids towards the device. Single males (*n* = 20) were left in silence for 20 min as control.

#### 2.3.2. Test 2. Female Playback on Pairs

In order to evaluate if the female playback could disrupt the mating behaviour, a pair (*n* = 30) consisting of a male and a female, was released on the cutting. Psyllids were placed on different leaves, which had approximately similar size (12–15 × 8 cm). In both treated and control groups, if a duet was not established after 5 min from the insect release, the signalling activity of the psyllids was triggered by playing the female playback. The pairs that, despite the use of playback stimulation, did not establish a duet were discarded. The tests ended after 30 min from the insect release.

#### 2.3.3. Test 3. Unspecific Noise Playbacks

Groups (*n* = 25) consisting of two females and two males were collected from the mass rearing and released on the same leaf of a cutting consisting of two leaves (approximate size: 10–12 × 8 cm). They were treated for 30 min with the transmission of two different playbacks, which were made using Audacity (Softonic International, Barcelona, Spain). Group 1, the “noise group”, was treated with a playback consisting of a sequence of 1.7 s long white noises with 0.3 s of silent intervals and frequency range between 0 and 22,000 Hz. Group 2, the “music group” was tested a playback obtained from the instrumental version of *Scary Monsters and Nice Sprites*. The song was created by music producer Skrillex and was used by Dieng and colleagues (2019) to disrupt the feeding and mating behaviour of the mosquito *A. aegypti* [26]. The track was downloaded from the Soundboard website (©2015 Soundboard, https://www.soundboard.com/sb/sound/937286) and then modified using Audacity. The song was cut from minute 1 to 3:40, and the selection was transmitted through the cutting as loop interspaced by 29.5 s of silence. The amplitude and the frequencies of both playbacks were chosen to cover the signals emitted from the psyllids on the leaf. In the case of the music playback, the stimulus could partially mask the psyllid signals because the song was composed of discrete frequencies and amplitudes. A control group (*n* = 25) was left in silence for 30 min. To assess whether the playbacks could decrease the psyllids’ signalling activity, we measured the number of males that called and those that established a duet with a female. We also counted the number of males that either achieved mating with a female or mounted another male, in order to determine whether the playbacks could interfere with the mating success and interspecific interactions, also in terms of partner’s misidentification. The number of psyllids that left the plant during the trials was used to evaluate if the playbacks could repel the insects from the host plant.

### 2.4. Data Analysis

In test 1, the call latency (i.e., the time from the release of the male to the first call) was compared between treatment and control using the Mann–Whitney two-sample test.

In test 2, a chi squared test in contingency tables (2 × 2), Bonferroni-corrected, was used to compare the number of searching males, the number of mating and the number of males that reached the mini-shaker. The Mann–Whitney two-sample test was applied to compare the time of the search latency (i.e., from the start of the trial to the onset of the male search), the time males took to reach the female, and the time took males to arrive at the node between the leaves. The same test was used for the comparison of the number of times the male visited the female leaf, the number of times the male visited the other two leaves, and the number of times a male moved down the stem.

In test 3, the Kruskal–Wallis test, followed by a Mann–Whitney pairwise test, Bonferroni-adjusted, was used to compare the duration of the copulations between groups. The number of calling males between each group was compared using a chi test in contingency tables (2 × 2), Bonferroni-corrected. A G test in contingency tables (2 × 3), Bonferroni-corrected, was applied to compare the male mating success, the frequency of male-on-male mounting and the number of insects that jumped off the leaf.

## 3. Results

### 3.1. Test 1. Female Playback Tested on Single Males

Although the number of calling males was rather similar between the treated and the control groups (11 and 8, respectively, *n* = 20), the call latency was significantly lower when the males were exposed to the playback (Mann–Whitney test: *n* = 8, 11; U = 1; *p* < 0.001). Within the treated group, nine of the calling males engaged in a duet with the stimulus and started to search for the source of the signal, reaching the node (*n* = 4) of the cutting and therefore getting close to the mini shaker. None of the control males moved from the leaf lamina after their release, regardless of their calling activity.

### 3.2. Test 2. Female Playback Tested on Pairs

In the control and treated groups, 19 and 9 males, respectively, called spontaneously thus triggering the female response and the duet. In the other cases (11 and 21, respectively), the onset of a male–female duet was triggered using the female playback. The duet was established if the male emitted a MCS and the female replied with a FRS. Most females (72.7% in the control and 66.6% in the treated group) replied to the vibrational stimulus before the males, a behaviour never reported before in *B. cockerelli* [23]. Females emitted their signals both during and after the end of the playback, while males always called after a female signal, which could be either artificial (playback) or natural (female psyllid). In both control and treatment, duets were established even if the female replied to the playback before the male. Throughout the test, females in the treated group overlapped their signal with both the male and the playback. Several females (9 out of 30) ceased to reply before the male could reach them or before the end of the trial. Similarly, 8 females in the control renounced to duet with the male prior to the end of the test.

Although the number of searching males was similar between the two groups (Chi squared 2 × 2 in contingency tables, *p* = 1; Table 1) and the male search latency was only slightly different (Mann–Whitney two-sample tests, *p* = 0.07; Table 2), the number of matings was significantly lower in the treated group (23.3%) than in the control (56.7%) (Chi squared 2 × 2 in contingency tables, *p* ≤ 0.001; Table 1). However, the time a male took to reach a female was similar between the groups (Mann–Whitney two-sample tests, *p* = 0. 8; Table 2), and mating occurred as soon as the male met the partner. Nonetheless, in one trial a treated male continued to call and searched for the female even after he had reached her on the plant. Similarly, the time needed for a male to arrive at the node between the leaves, once he had started to search, was similar between groups (Mann–Whitney two-sample tests, *p* = 0.8; Table 2). In both groups, the searching males visited the three different leaves several times (the one where he was released, the one that hosted the female and the empty leaf). Males moved from the node to the petioles and back, rarely walking into the leaf laminas. The number of times the male visited the female leaf (Mann–Whitney two-sample tests, *p* = 0.19; Table 2) was rather similar between the two groups, while the number of times the male visited the other leaves was significantly higher in the treatment than in the control (Mann–Whitney two-sample tests, *p* = 0.01; Table 2). The number of times a male chose to move towards the stem was also significantly higher in the treated group (mean ± SD 1.6 ± 1.8; *n* = 27) (Mann–Whitney two-sample tests, *p* = 0.002; Table 2), as well as the number of males that reached the mini shaker (50%; *n* = 30) (chi squared 2 × 2 in contingency tables, *p* = 0.006; Table 1). In the treated group, males (26.7%; *n* = 30) climbed on the device and remained on the device or on the stem in close proximity to it until the end of the trial.

### 3.3. Test 3. Disturbance Playbacks

The number of calling males was significantly different between the two treatments (Chi-squared test: χ^2^ = 22.2, df = 1, *p* ≤ 0.001). In particular, in the “noise group” only one male called, while in the “music group” more than 50% of the males emitted at least one calling signal. The number of calling males in the control was lower (40%), but not significantly different from the music group (Chi-squared: χ^2^ = 3.9, df = 1, *p* = 0.26). The difference was significant when comparing the control to the “noise group” (chi-squared: χ^2^ = 9.44, df = 1, *p* ≤ 0.02). Most of the calling males (58.8%) in the “music group” emitted their signals only during the silence gap, while the remaining 41.2% called both during silence and playback. The male call triggered either female response signals (1 FRS in the “music group”; 2 FRSs in the control group) or calling signals of the other male (3 MCSs in the “music group”). None of the females or males replied to the male call in the “noise group”. In case of a female response, a male–female duet was always established but led to mating only in one case (in the control group). Although female replies and therefore vibrational duets were sporadic, males could achieve mating in all the tested groups without signalling. The male mating success was similar between groups (G test: G = 0.3, df = 2, *p* = 0.8) since 40% (“music”), 36% (“noise”) and 44% (control) of the males eventually copulated with a female. There was no significant difference in the duration of the copulations between groups (Kruskal–Wallis test: χ^2^ = 2.9, df = 2; *p* = 0.2) and only one (“music group”) and three (control) male attempts failed due to female rejection. Male on male mounting was observed as well, with a similar frequency between groups (G test: G = 3.5, df = 2, *p* = 0.2), which was of 20% (“music”), 8% (“noise”) and 28% (control) for each group, respectively. Psyllids tended to walk along the leaf but remained on the cutting independent of the transmission of the playback, since the number of insects that jumped off the leaf was similar between groups (G test: G = 0.2, df = 2, *p* = 0.9).

## 4. Discussion

The present work demonstrated that the behaviour of *B. cockerelli* is vulnerable to manipulation by means of vibrational signals. In particular, the use of species-specific signals could be potentially used to interfere with the mating behaviour, either as a masking noise or an attractive stimulus. On the contrary, unspecific noises did not have any relevant effect on psyllids’ sexual behaviour.

The female playback successfully mimicked a *B. cockerelli* female, since it elicited the male search similarly to the actual female response in the test with pairs. Moreover, the playback prevented the male from visiting the female leaf and led him toward the device, resulting in a lower number of mating events. Although the mini shaker was in direct contact with the stem, *B. cockerelli* males frequently moved along the leaves and reverted their path back towards the node before reaching the leaf lamina. One hypothesis is that the directional cues provided by the playback could keep the male in between the petiole of the leaves and the node. In fact, the intensity of vibrational signals is very variable according to the plant architecture, and can be higher at certain transition or bifurcation points than in other parts of the plant [28,29,30]. Another possibility is that a real female, at short distance, could provide additional stimuli (i.e., chemical and visual) to searching males, and their absence could explain the tendency of males to not go directly to the source of the vibrational stimulus. Indeed, positive phototaxis and negative geotaxis also guided the male towards the top of the plant, where the actual female was placed, as hypothesised for *D. citri* [31]. In fact, *B. cockerelli* tends to move towards light stimuli, and UV-illumination was recently applied to enhance the attractiveness of sticky traps [32]. Notwithstanding, even in the presence of a replying female and natural light, our female playback prevented males from reaching the top of the plant and led them in the opposite direction. Moreover, when a male reached the stem (i.e., close to the mini shaker), he persisted in his search and often mounted on the device without any further reversion of his orientation. On the other hand, whenever the male entered the petiole of the female leaf, he did not turn backwards, thus ignoring the playback. Hence, when the male was on the “right” path towards a potential replying partner, he did not change direction. As a result, it would be very important to investigate the level of search accuracy of *B. cockerelli* males. In fact, this phenomenon could be even more relevant whether the male discriminates in favour of high-quality females and assesses the quality of replying females through the perception of their signals [33]. As an example, the accuracy of detection of a potential mate has been demonstrated for the planthopper *Hyalesthes obsoletus* when two different signals were simultaneously transmitted. In this species, the male distinguished perfectly between two different female signals with different frequency patterns and always decided to move towards one of them, which was specifically designed to contain more attractive features than the other [34]. Similarly, the importance of signals’ spectral features in affecting the male responsiveness to a female playback was also indicated in a study of the sharpshooter *H. vitripennis* [35], in which the male preferred synthetic playbacks endowed of certain spectral and temporal parameters. If *B. cockerelli* males can also discriminate between different female signals, then the spectral characteristics of the female playback could be modified to increase the signal attractiveness and lead more males towards the device, even in presence of other replying females on the plant.

Notably, most of the *B. cockerelli* females responded to the playback as well, and often their replies occurred before the male’s, which were always emitted after female signals. On the other hand, females signalled both during and after the playback, possibly as a form of rivalry behaviour to mask the vibrational cues provided by the “synthetic rival”. Although female insects are typically less competitive than males [33], some examples can be found in Pentatomidae [36], in the dance fly *Empis borealis* (Diptera) [37], and in some orthopteran and hemipteran species [33,38]. As for psyllids, female competition seems to be an original behaviour of *B. cockerelli*. In fact, *D. citri* females ceased to reply to the male when treated with the playback [7]. An alternative hypothesis to the intrasexual competition, could be that the *B. cockerelli* females tried to form a cooperative chorus with the playback to attract the male. However, since the device and the female were placed in opposite parts of the plant (the bottom and the top, respectively), their signals were competing to convince the males to take a directional decision and thus make a mating choice. On the other hand, both hypotheses (competition and cooperation) can coexist. According to this, a male could be more stimulated to search from the presence of multiple singing females, while females could establish a chorus as a means of decreasing the risk of being located by eavesdropping predators [39,40]. In this case, by overlapping her signal with the rival ones, the female would compete with the rivals in a relatively “safer” way. Besides cooperation or rivalry, the establishment of a chorus could be due either to an excess of sexually receptive females [33] or to female-biased gender ratio [36]. Furthermore, we must remember that psyllid populations can be rather dense and the contemporary presence of more males and females on the same plant is not rare. Nonetheless, there are no reports of intra-sexual rivalry within Psylloidea [41]. The role of female competition in *B. cockerelli* should be further investigated, since similar information could be used to improve a potential behavioural manipulation strategy. For instance, a highly competitive female playback could lead female to cease signalling, as in the case of *D. citri* [7].

The second approach that we tested to interfere with the *B. cockerelli* behaviour was the use of unspecific vibrations. It is known that cerambycids, for instance, respond to generic vibrational signals [4,42], and some of their behaviours such as feeding, walking and oviposition can be disrupted by the transmission of artificial vibrations [4]. In Coleoptera, immobility seems to serve as a strategy to recognise approaching conspecifics or predators [4]. In the Japanese pine sawyer *Monochamus alternatus*, for example, vibrations below 1 kHz induced startle and freeze responses [4]. Notwithstanding, none of our disturbance playbacks triggered similar responses in *B. cockerelli*. Neither the walking activity nor the physical and vibrational interactions of the psyllids were affected during the playback emissions. However, the possible interference with other behaviours such as feeding and oviposition cannot be excluded. Besides freezing responses, animals can respond to disturbance noises by modulating their signalling activity and/or leaving the treated area. In some anuran species, for instance, anthropogenic noise affected the calling rate and the spatial displacement of the frogs, which moved away from the source of the disturbance [43]. Although none of our playbacks induced the psyllids to leave the host plant, the males differently adjusted their signalling behaviour in response to either the noise or music. In fact, while the male calling activity was significantly reduced when broadband noises were transmitted through the plant, males did not cease to call when treated with electronic music. Although habituation is more likely to occur when animals are treated with repetitive noises [44], insects commonly avoid signalling in the presence of disturbance sounds and wait for silence [45]. Thus, the short silent gap between the sequences of white noises (less than 0.5 s) likely discouraged *B. cockerelli* males from signalling, while the calling males in the music group exploited the long silent gap (30 s) within the song playback to start calling. Indeed, besides the temporal pattern, the disruptive potential of a playback can also be determined by spectral features. In the fruit fly *Drosophila montana*, for instance, the transmission at high intensities of a background noise with similar frequency to the male courtship signal resulted in a decrease of both female replies and male mating success [46]. Similarly, the most efficient disturbance noise to disrupt mating in *Empoasca vitis* was a playback set on the dominant frequency of the male call [47]. In *H. vitripennis*, the transmission of a noise with the dominant frequency of the female call (80 Hz) reduced the male replies and, consequently, the mating [35]. The music playback did not prevent *B. cockerelli* males from calling, probably because the song consisted of a mixture of sounds with discrete and rational frequencies, which were less efficient in masking the spectral contents of psyllids’ calls than the broadband noise. In fact, although males tended to start calling during the silent gap, some individuals emitted their signal during the music playback. Conversely, the sequence of noises completely covered the frequency range of *B. cockerelli*’s signals, which are composed of broadband pulses ranging from 190 to 1125 Hz with the dominant at 548.5 ± 257.6 Hz [23]. Although this disturbance noise could be deployed to suppress the male calling activity in crops, the broad frequency range could also affect other insect species (pest and/or beneficial insects) responding to vibrational signals and present on the plant. Accordingly, only the frequencies that are in the sensitivity range of insect vibro-receptors would have an impact on the pest’s behaviour and communication. In insects, the detection of vibrations is mainly achieved by chordotonal organs and among these, the subgenual organ is the most sensitive and responds to a frequency range between 50 and 1000 Hz [48]. Consequently, the signals used to communicate must also lie in the same frequency range. The noise playback, with its wide frequency range would mask these signals and interfere with intra-specific communication of other insects than *B. cockerelli*. As example, green lacewings (Chrysopidae: Neuroptera) are responsive to vibrations and use substrate-borne signals to achieve mating [49,50]. These insects are effective predators and feed on many important agricultural pests, and for these reasons they are used in biological pest control [51].

Therefore, more tests are needed to both assess how background signals could affect the vibrational communication of useful insects in the field, and better understand which features best comply with the *B. cockerelli* behaviour to design species-specific signals to be used for population control. Besides mating disruption, disturbance vibrational signals could be used to disrupt the feeding behaviour of the psyllids, as has been reported for cerambycids [4]. It would be therefore worthwhile to evaluate whether the playback can interfere with feeding by using an Electrical Penetration Graph (EPG) technique during the emission of the stimulus [52].

A control strategy against *B. cockerelli* based on the use of disturbance vibrations should be applied as a tool within a wider strategy of IPM to keep the infestations low. At high population density the efficacy of the method would be probably lower, as suggested by the fact that *B. cockerelli* pairs occurring in close vicinity (in our trials we released four psyllids on the same leaf), managed to mate. Therefore, the mating success due to causal encounters was not negatively affected by the disturbance playbacks. In this regard, we hypothesise that vibrational signals are not always necessary to achieve mating, since the “silent” copulations seem to be common in *B. cockerelli* as well as in other Psylloidea. In fact, in some species a male can approach the female and successfully achieve copulation without any preliminary signals [53,54,55]. In other hemipterans such as leafhoppers (Cicadomorpha), pair formation is strictly dependent on the exchange of different vibrational signals that are associated to different phases of the mating process (i.e., identification, localisation, courtship) [56,57]. Unlike leafhoppers, the psyllids’ repertoire of vibrational signals is typically limited to the male call and female reply, which are used invariantly during pair formation [20,53]. This can be explained by the fact that psyllids can live in dense colonies on the plant [58] and, in these circumstances, intra-specific interactions can minimally rely on exchanges of vibrational signals. In fact, we observed frequent mating and male-male mountings when both males and females were released on the same leaf in test 3. Same-sex sexual interactions are probably a result of high sexual responsiveness and could be explained as mistaken identification by the mounting male. As suggested for many other arthropods, the benefit of attempting mating with a conspecific (regardless of the gender) exceeds the costs of a misidentification [59]. Although it is possible that there is an adaptive explanation for same-sex sexual interactions, unveiling the role of this behaviour in *B. cockerelli* was not the aim of this work. Nonetheless, future research should be addressed in this regard.

Further studies should also assess if other signal modalities are involved in the mating process and that can be synergic or complementary to vibrations to accomplish important tasks (i.e., directionality, mate choice, female acceptance) [21]. Accordingly, the role of odours in the mating behaviour of *B. cockerelli* has been demonstrated by Guédot and colleagues [60]. In their experiments, males were attracted by both live females and female odorants, which are likely sex attractants. Since the researchers did not evaluate whether the psyllids were also emitting vibrations, further trials should assess the use of multiple cues in the mating behaviour of *B. cockerelli*. If pair formation depends on both vibrational and chemical signals, these stimuli could be coupled in order to design a future technique to trap the insects and/or disrupt their mating behaviour. Accordingly, an example of manipulation of psyllids’ behaviour by using multiple cues has been recently proposed for *Cacopsylla picta,* an important pest of apple [55]. In fact, the sexual communication of the genus *Cacopsylla* seems to rely on both vibrational and chemicals signals, and plant volatiles play a role as well [55,61,62].

## 5. Conclusions

To summarise, our study demonstrated that the mating behaviour of *B. cockerelli* can be affected by the transmission of a female playback, when a female and a male occur on different leaves. Furthermore, the broadband playback negatively affected the psyllids’ calling behaviour. On the other hand, the disturbance test also demonstrated that *B. cockerelli* pairs that occur on the same leaf can meet and copulate even in the presence of background noises. Therefore, a vibrational mating disruption approach is likely to be efficient if applied before high rates of infestations are reached and when, therefore, the partner location can depend on vibrational communication. A remarkable outcome is that females can establish choruses, a phenomenon that was never reported in *B. cockerelli*, although at this stage of research, their role is not yet clear.

Indeed, even if vibrational manipulation of *B. cockerelli*’s behaviour has the potential to become a tool for an integrated pest management approach, technical constraints must be considered before the method can be used in the field. Although disruptive signals for the control of *S. titanus* were transmitted to grapevine plants by exploiting the wires in the vineyard [6], a similar strategy may be not be suitable for potato crops. A possible option could be the induction of vibrations in host plants by means of sound speakers. For instance, mating was disrupted in *Amrasca devastans* and *Nilaparvata lugens* by aerially transmitting disturbance sounds of 80 dB, 200–300-Hz [63]. Similar approaches should therefore be tested in field environment for a future use of the female playback and other disturbance signals to disrupt the mating behaviour in *B. cockerelli*.

## Figures and Tables

**Figure 1 insects-11-00299-f001:**
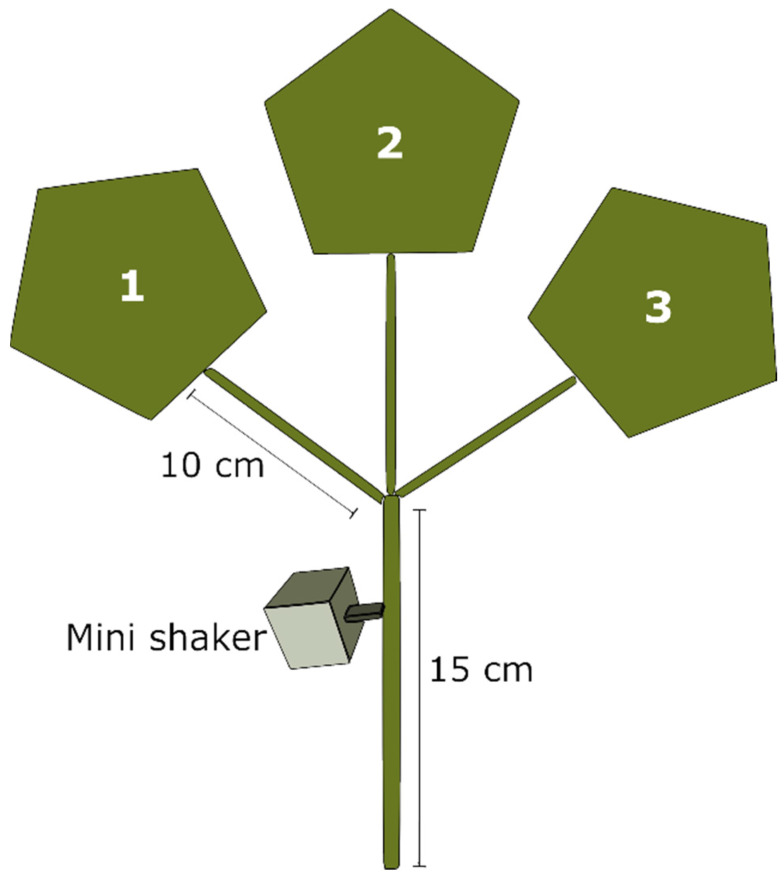
Schematic drawing of the experimental setup with a capsicum cutting, which consisted of either two (test 1 and 3) or three (test 2) leaves. The mini shaker was always pointed 2–3 cm below the node.

**Table 1 insects-11-00299-t001:** Results of the test chi squared 2 × 2 in contingency tables for control and treatment (female playback) in Test 2. The test was used to evaluate if the female playback could decrease the male mating success (Number of mating) and attract the male towards the source of the stimulus (mini shaker). Number of male–female pairs of *Bactericera cockerelli* tested per each group = 30.

Measured Parameters	Number of Males Analysed for Each Parameter		χ^2^	df	*p*
	Control	Playback			
Number of searching males	27	26	0.16	1	1
Number of mating	17	7	6.94	1	≤0.001
Number of males that reached the mini shaker	0	8	7.5	1	0.006

**Table 2 insects-11-00299-t002:** Results of the test Mann–Whitney two-sample tests for control and treatment (female playback) in Test 2. The test was used to compare the time spent by the male to search and to reach the female and the node, and to compare the number of times he visited the different leaves and the stem. Number of male–female pairs of *Bactericera cockerelli* tested per each group = 30.

Measured Parameters	Number of Males Analysed for Each Parameter		Median		U	*p*
	Control	Playback	Control	Playback		
Start of male search	27	26	180 s	365 s	250	0.07
Time to reach the female	17	7	980 s	960 s	56.5	0.8
Time to reach the node	17	7	660 s	712 s	56.5	0.8
Number of times the male visited the female leaf	25	27	1	1	274	0.19
Number of times the male visited other leaves	25	27	1	2	213	0.01
Number of times the male went towards the stem	25	27	0	1	195.5	0.002

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
