# Peer review of "Can Vibrational Playbacks Disrupt Mating or Influence Other Relevant Behaviours in Bactericera cockerelli (Triozidae: Hemiptera)?"

_insects, 2020, doi:10.3390/insects11050299_

Round 1

Reviewer 1 Report

In this paper, Avosani et al. presented experimental evidence for vibrational playbacks in mating of the tomato potato psyllid Bactericera cockerelli as a potential for pest management techniques using vibrations. This is a well-written paper with deliberate discussions and a very good contribution to the special issue of Insects. In order to strengthen the significance of their findings, I would suggest they modify minor points listed below.

L20: Delete unnecessary 'either'.

L58-60: Here it is necessary to mention male calling song (MCS). In line of 183, abbreviation of MCS is suddenly described without explanations.

L119,120: I am sure that you monitored intensities or velocities of playbacks on the plant in addition to temporal and spectral values. It is better to include such information on the intensities briefly.

L175,220,224: Delete unassay asterisks.

L189: Replace several or another word from 'Few'.

L197,201: I think the p values of one figure (0.08) is incorrect. The values are 0.8 in Table 2.

L208,209: Please include percentages for the contingency table in N of times the male went towards the stem.

L214: Delete unnecessary 'p =' from Table 1

L218: It is kind for readers to include medians or other related values to show differences by Mann-Whitney U tests in Table 2. You may describe them in the text, instead of the table.

L219: Please include percentages of the two treatments.

L356: It may be appropriate to include references regarding EPG.

L391,442,etc: Genus and species should be italic and species should be small letters.

L539: Change Zool from Zoolog

Reviewer 2 Report

I detected only a few small mistakes. First, I found only in the literature section a missed space (line 524, paper 36., flyEmpis) and another mistake at the end of ms (line 391) Cacopsylla instead of Cacopsylla.

Reviewer 3 Report

This manuscript covers important aspects of Bactericera cockerelli vibrational communication that may be of value for management of this important pest insect. Several revisions are needed but most of them are minor. The two tables are difficult to interpret, and the reviewer suggests changes at the following lines:

Line 58, 489 This reference is incomplete, and the reviewer could not find.

Line 103 “was used to control”

Line 109 The reviewer did not find a listing of the signal amplitude produced by the minishaker. Presumably it was set to be similar to the amplitudes of the psyllids or was some known multiple of the psyllid signal amplitude.

Line 170 The reviewer had trouble interpreting the tables. It may be helpful if the authors listed the null hypotheses tested.

Line 183 It is not clear why this was unexpected. Perhaps the reason is explained in reference 23 and a summary description could be given here in the methods section or introduction. Certainly, in other psyllid species, the male call or female reply often is not directly triggered by a female signal. In some species, however, the female almost always replies immediately after the male call.

Line 197 ? should this be p = 0.8? or is the table incorrect?

Line 209 There may be an error in the number of males that reached the minishaker in Table 1. Based on the statement that 26.7% of 30 climbed on the device, N should be 8 for the playback. To obtain a chi square of 7.5, the number in the control would be < 2, the reviewer suspects.

Line 212 The column headings for the table are confusing. For “Measurement” do the authors mean “Behavior Measured”. For “Number of observations” do the authors mean Number measured per group?

A similar problem of wording may also be occurring in Table 2.

It is confusing why U is different for the last three measurements in Table 2 when the numbers of observations for each type of measurement are identical. Possibly the reviewer is misunderstanding what is the null hypothesis or what the numbers in each column mean.

Line 254 The text location for Reference 30 is missing but it may go with 29 [28-30] or [29-30]? [28] seems more likely to go on line 255 as “additional stimuli (i.e., chemical [28] and visual)”

Line 285 Note: heteroptera is no longer considered a suborder so the term should be hemipteran

Line 343 “responses” should be “responds”
